# Intensification of Heat and Mass Transfer in a Diabatic Column with Vortex Trays

Nikolai A. Voinov, Anastasiya V. Bogatkova and Denis A. Zemtsov *

Institute of Chemical Technology, Reshetnev Siberian State University of Science and Technology, 31 Krasnoyarsky Rabochy Av., 660037 Krasnoyarsk, Russia; n.a.voynov@mail.ru (N.A.V.); sonchic.sveta@yandex.ru (A.V.B.)
* Correspondence: denis_zemtsov.92@mail.ru

**Abstract:** We used vortex contact devices that we developed and investigated to make a new design of an alcohol diabatic distillation column with heat exchange pipes (as the reflux condenser) passing through concentrating section trays. In the column, ascending vapors partially condensed on the surface of vertically installed heat exchange tubes, forming a reflux. The reflux was then mixed with the draining liquid flow in the vortex contact devices placed on the trays. Heat was removed from the column through the boiling of the draining water film along the inner surface of the heat exchange pipes. We compared both diabatic and adiabatic columns fitted with the developed vortex contact devices on the trays. The proposed innovative contact system allows increasing productivity, reducing column dimensions and steam- and heat-transfer medium consumption, and increasing separation efficiency. Dependences for calculating the gas content, hydraulic resistance, and interphase surface required for designing the vortex contact devices of the proposed unit trays are presented.

**Keywords:** vortex devices; contact stages; efficiency; diabatic rectification; hydrodynamics; heat-exchange; mass transfer

## 1. Introduction

Rectifying columns are widely used in industry to separate mixtures and make products for different purposes. The adiabatic units that are currently used have high energy consumption, which results in increased rectification costs. Rectifying columns have a height often exceeding 25 m. All this leads to high costs related to the manufacture, operation, and repair of rectifying columns.

The method used to organize refluxes in the adiabatic units does not provide optimal distribution of mixture components both on the trays and along the column height. The steam flow passes through the column and is condensed in the heat exchangers (reflux condenser). The resulting condensate enters the upper column trays with a maximum flow. This leads to several factors that reduce unit efficiency. In particular, with this reflux formation, the intensity of vapor condensation on the surface of heat exchanger pipes decreases due to the relatively low thermophysical parameters of the condensed vapors produced in the upper part of the adiabatic column (low phase transition criterion). This leads to an unjustified increase in the heat exchange surface, the dimensions of the heat exchange devices, high metal consumption, and increased consumption of heat-transfer media. Reuse of the heat-transfer medium passed through the reflux condenser requires cooling, which also results in additional energy consumption.

The heat and mass exchange processes (condensation and evaporation) on the trays along the column height are not synchronized between the mixture low-boiling and high-boiling components due to the difference in the physical properties of the draining maximum reflux flow and rising vapors. This suggests the practicability of creating a reflux on each tray, for example, by partial condensation of vapors rising along the column height.

Additionally, in the adiabatic column, we observe the supercooling of the reflux flow on the surface of multi-pass heat exchangers, despite the constant improvement of the

reflux condenser design. It is obvious that a decrease in the heat-transfer medium flow rate triggers a laminar flow in the pipes, which is not permissible, and its increase leads to the pipe surface and condensate cooling. Use of different heat exchange intensifiers in multi-pass reflux condensers leads to an increase in dimensions and metal consumption. Reflux supercooling requires additional heating to the boiling point. The reflux not heated to the boiling point condenses volatile components (impurities) on the upper trays and entrains them to the bottom of the column. It is therefore not possible to completely separate and purify a distillate from impurities, which is currently in demand.

An unreasonably large flow of the reflux on the upper trays leads to an increase in the diameter of the concentrating section of the column to accommodate overall overflows on them and increase the number of contact devices.

The displacement of a large vapor flow along the column results in increased unit resistance and heat losses.

The use of conventional contact devices on trays such as fixed and movable valves or caps is justified only at low unit capacity. An increase in the liquid flow rate leads to a sharp decrease in the tray efficiency due to the flow of gas bubbles from the gas–liquid layer. Additionally, an increase in the steam load on the trays with such contact devices leads to liquid pulsations, drop entrainment, and flooding. Stagnant zones form on the trays, which leads to the formation of deposits on their surfaces.

With conventional trays, a large volume of liquid is typically placed to provide a certain height for the liquid layer, which keeps them from being used, for example, for processing explosive and thermolabile mixtures. This also keeps the unit from being quickly brought to its operating mode.

The above factors prove that new design solutions are required both in the creation of rectifying columns and in the choice of contact devices on the tray.

One method for intensifying the separation process is through managing the process in diabatic rectification and using vortex contact devices on trays.

Diabatic distillation is a successful example of a simple process intensification technology for improving separation and thermal efficiency. To convert an adiabatic column into a diabatic one, a reboiler and condenser are partially replaced with heat exchangers built into the column [1,2].

Replacing an adiabatic unit with a diabatic one allows reducing the temperature difference between the heater and the cooling liquid [3]. This process uses cheaper heat-transfer fluids and coolants.

In turn in terms of their characteristics, centrifugal mass-exchange contact devices are most suitable for solving the problem of reducing equipment dimensions and increasing the productivity and efficiency of existing mass-exchange units. Centrifugal mass-exchange contact devices are able to provide up to 80% higher efficiency than traditional units [4,5].

The most advanced achievements in this area today are centrifugal trays such as Swirltube, ConSep Sulzer [6–10], Ultra-Frac [11,12], CoFlo [13,14], and UOP SimulFlow [15].

Different centrifugal tray designs are conventionally divided into vortex and film direct-flow vortex units. In film direct-flow vortex contact units (see Figure 1), steam (gas) transports liquids as a film along the inner surface of pipes mounted on trays [16–18].

They are notable for their high gas capacity. However, due to their high metal content, low liquid output, and the presence of stagnant zones on the tray deck, they are not widely used.

The designs of vortex trays with tangential swirlers in contact devices (Figure 2, [19,20]) are easy to manufacture and provide high efficiency, but require structural refinement.

We conducted our first diabatic rectification studies for a column 0.1 m in diameter with 25 vortex trays (see Figure 2a), which were fitted with dephlegmators in the form of coils [21] (see Figure 3a).

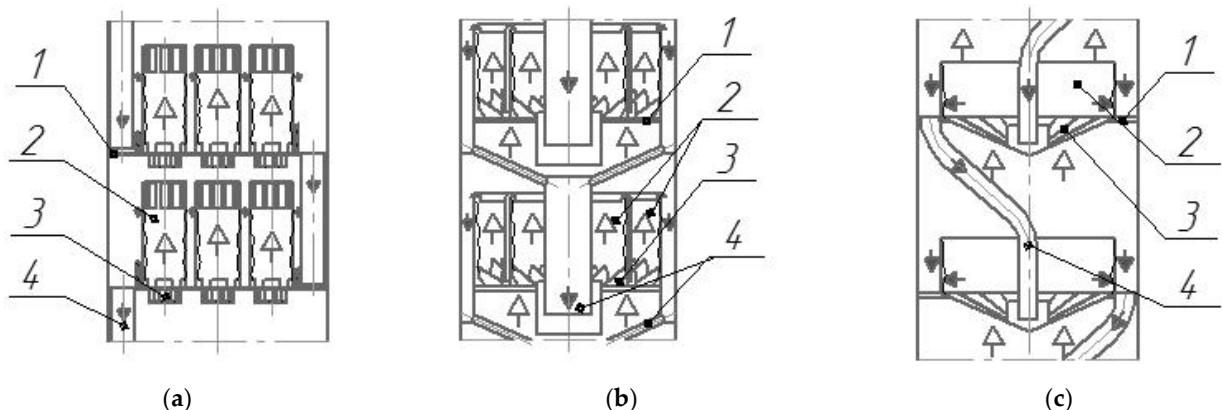

**Figure 1.** Diagrams of direct-flow vortex trays with tangential swirlers (**a**) and axial ones (**b**,**c**). 1—tray deck; 2—vortex contact device; 3—swirler, 4—overflow device; ►—liquid; ▷—steam.

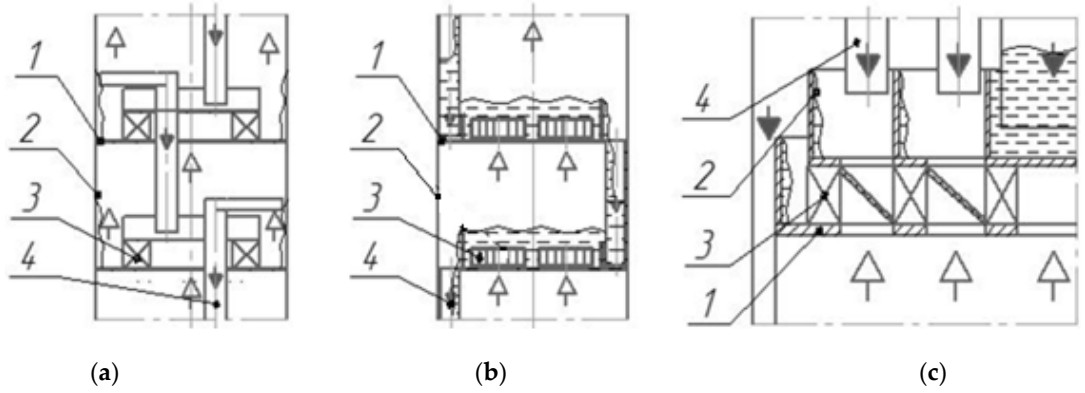

**Figure 2.** Diagrams of vortex trays with tangential swirlers: vortex chamber (**a**,**c**), vortex tray (**b**). 1—tray deck; 2—shell; 3—swirler; 4—overflow device; ►—liquid; ▷—steam.

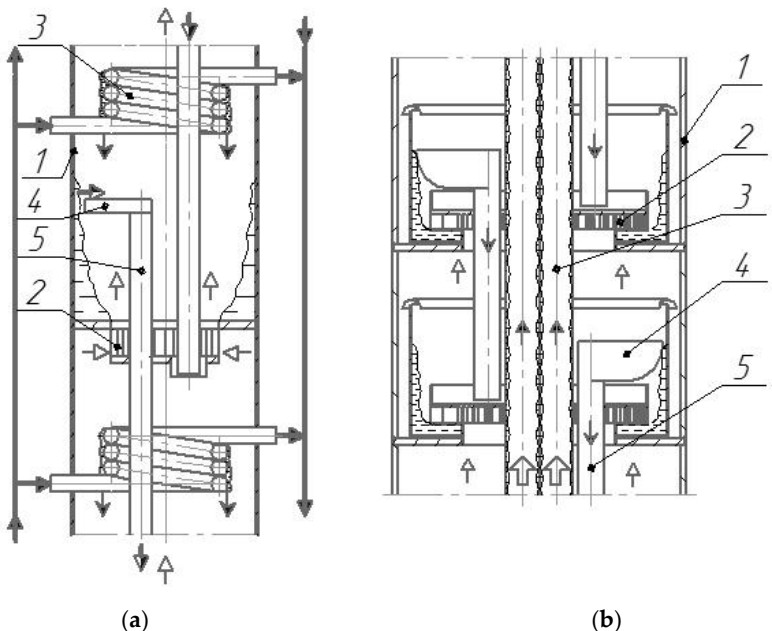

**Figure 3.** Diagrams of the built-in dephlegmators in the column in the form of coils (**a**) and pipes (**b**). 1—housing, 2—swirler, 3—dephlegmator, 4—drain bar, 5—overflow device; ►—liquid; ▷—steam.

According to the average data obtained on the column, the tray efficiency in diabatic rectification was 0.8 and in adiabatic rectification 0.4. We obtained the dependency to determine the stage efficiency as follows [21]

$$E = 0.035 \times m^{0.16} \times (G/L)^{-0.15} \times (H/h)^{0.4} \times Re^{0.24}, \tag{1}$$

where $E$ is the Murphree efficiency; $m$ is the slope of the equilibrium curve of the mixture; $G$ is the vapor phase flow rate, kg/s; $L$ is the liquid phase flow rate, kg/s; $H$ is the liquid level at the stage, m; $h$ is the swirler channel height, m; $Re$ is the Reynolds number.

It has been shown that supplying the heat-transfer medium to the dephlegmators placed on the trays has a major influence on the separation process in the diabatic column.

In this regard, a promising heat exchange device for carrying out partial condensation of vapors on the diabatic column trays is a dephlegmator made of pipes vertically mounted along the column height [22,23] (see Figure 3b).

In the dephlegmator made of pipes (see Figure 3b), it is possible to intensify vapor condensation and reduce condensate (reflux) supercooling. The dephlegmator is compact, easy to manufacture, and has a low metal content. The device also does not significantly affect the hydrodynamics and mass exchange on the trays. Temperature stresses are easily eliminated; for example, by installing a lens compensator on the pipe surface.

To intensify heat removal in the pipe dephlegmator, a film flow was placed over their surface [24]. At that, it is possible to ensure not only gravitational flow of the heat-transfer film, but also an ascending and descending film flow without significant changes in the design [25,26]. In this case, the heat-transfer coefficient is an order of magnitude higher than that for the single-phase flow in the heat exchanger pipes. Heat removal is carried out at a lower heat-transfer medium flow rate. Combined heat removal can be carried out in the dephlegmator pipes by evaporation from the film surface, heating, and boiling.

For example, maintaining the boiling process in the draining film allows intensifying heat transfer and maintaining the same temperature of the heat-transfer medium along the column height. The magnitude of the heat-transfer coefficient during the boiling of the gravitationally flowing water film reaches [27] 10,000 W/(m$^2$ K) and more.

This paper presents work on the development of a diabatic column design employing a new approach to reflux formation on trays and the organization of methods for intensifying heat removal and separation, which allows reducing capital and operating costs due to a decrease in unit dimensions and steam and heat-transfer medium consumption.

We developed new designs of contact vortex devices on trays, which allowed lowering the volume of liquid on the tray deck, reducing sediment formation conditions and the number of stagnant zones, and increasing tray efficiency and steam and liquid productivity.

## 2. Materials and Methods

### 2.1. Designs of the Experimental Rectification Unit and Vortex Devices

For the trays of the reinforcing rectification column, a vortex contact device [28], shown in Figure 4a, was developed, efficiently operating at relatively low liquid loads.

For exhaustive columns where high liquid loads are required, the devices [29] shown in Figure 4b,c were developed.

The contact device shown in Figure 4a consists of a cylindrical insert 1, swirler 2 equipped with bottom 3 and cover 4 where hydraulic lock cup 5, overflow device 6 with drain bar 7 and a sealing device 8 for heat exchange pipes (not shown in Figure 4a) are located. Bottom 3 has steam passage channels 9.

The bent part of the drain bar 7 forms a curvature of the rotating gas–liquid layer surface in the device and ensures liquid flow into the overflow device 6. The liquid is supplied via the overflow device to a cavity of hydraulic lock cup 5, and then, under the influence of centrifugal force, is pressed onto the surface of cylindrical insert 1. The gas passes through the channels of swirler 2 in rotational motion and contacts the gas–liquid layer rotating on the surface of cylindrical insert 1.

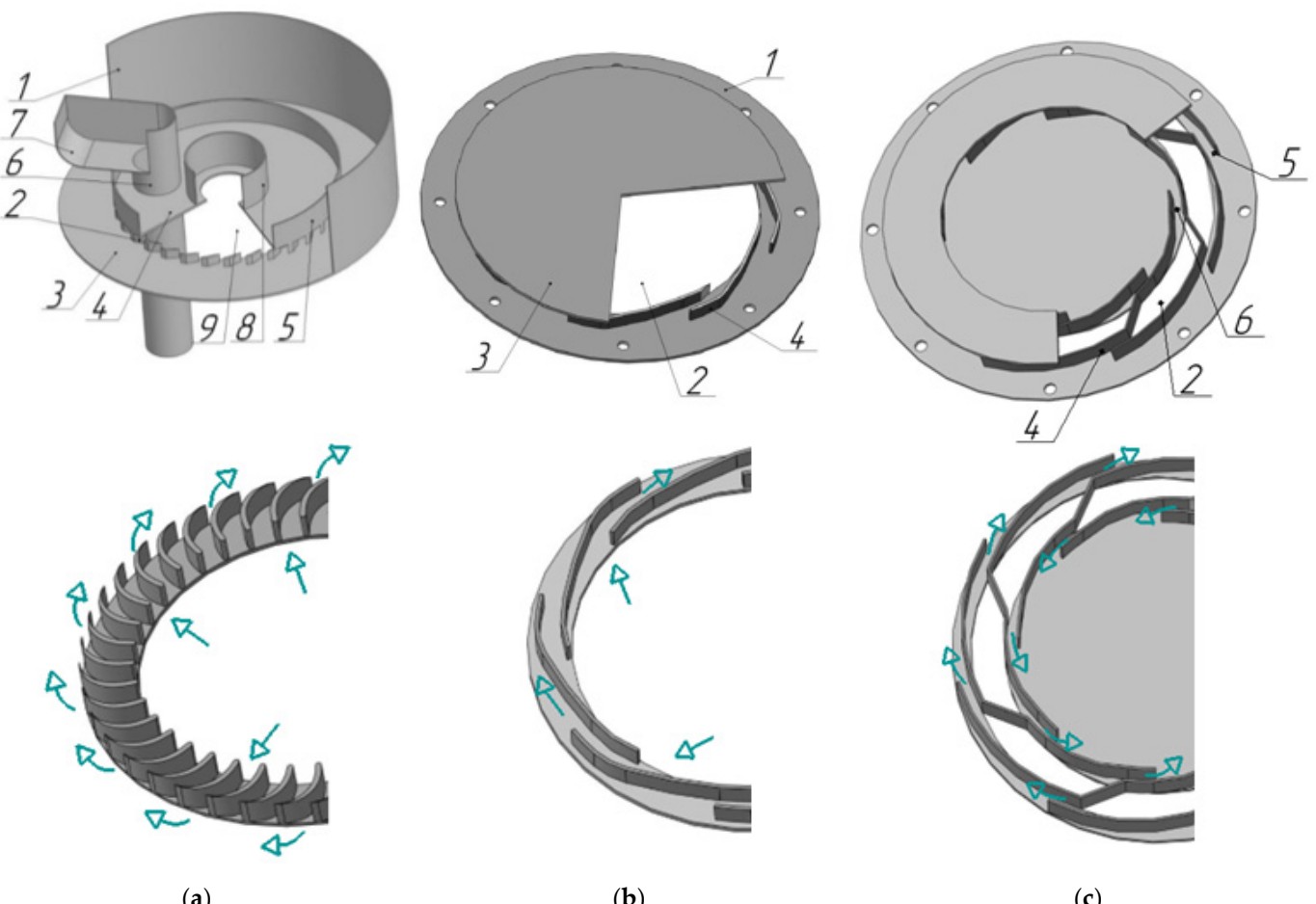

(**a**)  (**b**)  (**c**)

**Figure 4.** Diagrams of the vortex contact devices and steam movements in the swirlers with profiled channels (**a**) and annular channels (**b**,**c**). (**a**) 1—cylindrical insert 2—swirler; 3—bottom; 4—cover; 5—hydraulic lock cup; 6—overflow device; 7—drain bar; 8—sealing device; 9—steam channel. (**b**) and (**c**) 1—tray deck; 2—steam channel; 3—cover; 4—tangential swirler; 5 and 6—inner and outer annular swirler channels; ▷—gas.

The contact device (Figure 4b,c) consists of tray deck 1 fitted with gas passage pipe connector 2, cover 3, and tangential swirler 4 with annular channels 5 and 6.

Steam enters the device via channel 2 into the swirler. It acquires rotational motion and disperses in the form of currents into the liquid located on the tray deck, thereby forming a developed interphase surface [30].

The design parameters of the studied swirlers are as follows: channel height and width $h = 0.003$–$0.08$ m, $b = 0.0015$–$0.010$ m, channel wall slope angle $\alpha = 26°$. Outer swirler radius $R_{out} = 0.103$ m. Inner swirler radius $R_{in} = 0.054$–$0.092$ m. Number of channels $n = 8$–$40$ pcs. The length of the channel varied from 0.004 to 0.022 m.

The experimental unit designed to study the rectification process is presented in Figure 5.

The working volume of still 1 of the experimental unit was 0.1 m³. Electric heaters with total power of 30 kW were fitted in the still heat exchange jacket. Column shells 2 were 0.2 m in diameter, between which three trays 3 were fitted. Above the upper tray was a condenser 15 made of a copper pipe 6 mm in diameter in the form of a coil. Vortex contact devices 4 (also shown in Figure 4a) were mounted on the trays. A heat exchange pipe 5 in diameter 0.028 m was passed through them. The upper end of pipe 5 is equipped with heat-transfer medium (water) injection chamber 16. The lower end is connected to storage containers 9 connected by pipelines with vacuum pump 8.

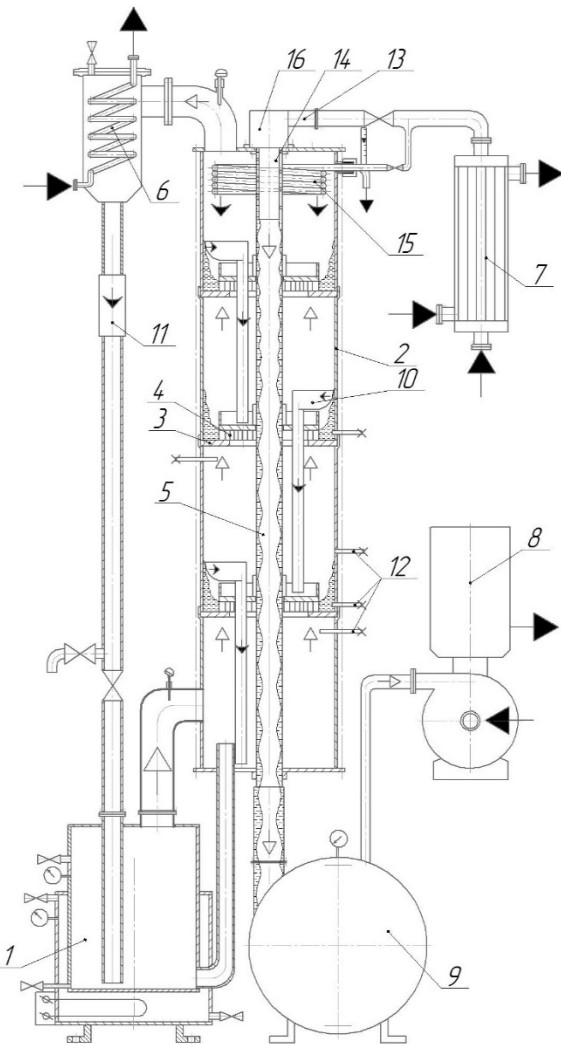

**Figure 5.** Experimental unit set-up. 1—still; 2—shell; 3—tray; 4—contact vortex device; 5—dephlegmator; 6—condenser; 7—heat exchanger; 8—vacuum pump; 9—storage containers; 10—drain bar with overflow device; 11—inspection window; 12—samplers; 13—connecting piece for heat-transfer medium supply; 14—liquid distributor; 15—upper tray condenser; 16—chamber for heat-transfer medium injection. ▶—heat-transfer medium; ▷—steam; ➷—reflux (condensate).

Vapors of the reinforced mixture condensed in condenser 6 are returned via the hydraulic seal to still 1.

For adiabatic rectification, the heat-transfer medium was fed only to condensers 15 and 6.

In studying diabatic rectification, the heat-transfer medium was heated in heat exchanger 7 to the required temperature and fed to dephlegmator pipe 5.

On the inner surface of pipe 5, liquid distributor 14 provided a draining liquid film. A vacuum created in container 9 provided for boiling of the draining water film and evaporation of moisture from the surface. This ensured the partial condensation of rising vapors in the unit and the formation of a reflux on each tray.

### 2.2. Experimental Data Processing

In studying the hydrodynamic parameters of the vortex devices, the flow rate of the supplied liquid was recorded using a rotameter and the gas flow rate using a diaphragm.

Differential pressure gauges were used to measure the tray pressure drop.

The resistance coefficient was determined according to

$$\xi = \frac{2 \times \Delta P}{u_G^2 \times \rho_G},$$ (2)

where $\Delta P$ is the swirler pressure drop, Pa; $u_G$ is the average gas flow rate in the swirler channels, m/s; $\rho_G$ is gas density, kg/m$^3$.

The critical gas velocity in the swirler channels and the angular velocity of the gas–liquid layer, were determined using video recording.

The magnitude of the gas content in the liquid layer was determined by the volumetric method according to

$$\varphi = \frac{H_{G-L} - H_0}{H_{G-L}},$$ (3)

where $H_0$ is the height of the liquid layer on the tray, m; $H_{G-L}$ is the height of the gas–liquid layer, m.

The interphase surface was determined as

$$a = \frac{6 \times \varphi}{D_b},$$ (4)

where $\varphi$ is gas content; $D_b$ is the average surface diameter of the bubble, m.

Dissipation of the gas energy $\varepsilon$ (W/kg) supplied into the working volume of the shell through air currents was calculated according to the formula

$$\varepsilon = \frac{U}{M},$$ (5)

where $U$ is internal gas energy, W; $M$ is liquid weight, kg.

The mass transfer on the vortex contact tray shown in Figure 4b,c was studied using the example of water absorption of air oxygen according to [31].

The initial concentration of oxygen in the water was $0.1 \times 10^{-3}$ kg/m$^3$. A polarographic sensor was used to measure the oxygen concentration in the water.

The water flow rate with reduced oxygen content ranged from 0.1 to 1.0 m$^3$/h.

The mass transfer intensity was determined according to the dependence:

$$\beta_V = \frac{Q_L(c_o - c)}{-V(c^* - c)},$$ (6)

where $Q_L$ is the flow rate of oxygen-free water supplied to the tray, m$^3$/s; $c_0$ is the concentration of oxygen in the liquid supplied to the tray, kg/m$^3$; $c$ is the concentration of oxygen in the liquid on the tray, kg/m$^3$; $V$ is the volume of liquid on the tray, m$^3$; $c^*$ is the equilibrium concentration of oxygen in the liquid, kg/m$^3$; $\beta_V$ is the volumetric mass transfer coefficient, s$^{-1}$.

Tray efficiency:

$$\eta = \frac{c - c_0}{c^* - c_0}.$$ (7)

The rectification and heat exchange process was studied this using the unit presented in Figure 5.

The vortex tray efficiency during rectification was determined according to the Murphree formula:

$$E = \frac{y_{ex} - y_{ent}}{y^* - y_{ent}},$$ (8)

where $y^*$ is the equilibrium concentration of vapors with the liquid on the tray, % mol; $y_{ent}$ is the ethanol vapor concentration at the tray inlet, % mol; $y_{ex}$ is the ethanol vapor concentration at the tray outlet, % mol.

In rectification, the initial concentration of the mixture in the still was (1–60)% wt. Ethanol concentration was measured in the vapor and liquid phases. The composition of ethyl alcohol was determined using a LR-3 refractometer and an alcohol meter calibrated used a YCDplus a YCDplus mass-spectrometer.

The temperature indicators in the liquid and vapor phases were determined using TSM-9418 resistance thermometers, secondary Thermodat 35TsO/GVS devices calibrated using glass thermometers with a scale division of 0.01 °C, taking into account the influence of the mixture vapor velocity on the temperature indicators at the sensor site.

The flow rate of steam and liquid in the column was calculated based on material and thermal balances.

### 3. Results and Discussion

*3.1. Hydrodynamics on Trays with Vortex Devices*

Resistance of Vortex Devices

Devices (swirlers) are designed to create a rotational motion for a gas–liquid flow on the tray. Swirlers with straight steam passage walls have a simple design and therefore have found widespread application in industry. However, as the channel width increases, there is need to increase the channel wall length to maintain the tangential component of flow rate, which results in an increase in the device's hydraulic resistance. In this regard, there are swirlers with profiled channel walls (see Figure 4a) that eliminate this drawback. At that, the latter have a relatively low resistance (Figure 6, profiled channels) and therefore have been accepted in the design of vortex devices located on the concentrating column trays.

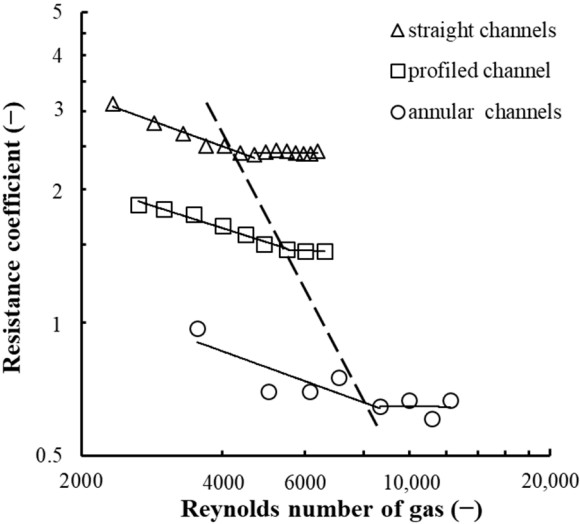

**Figure 6.** Dependence of the magnitude of the resistance coefficient on the gas Reynolds number. At $b = 0.0035$ m, $h = 0.008$ m, $n = 40$ pcs, $R_{out} = 0.085$ m, $l = 0.022$ m, $\alpha = 26°$; Experimental points (1–3): for different channel wall profiles.

The improved swirler design has led to the creation of devices with annular channels, as shown in Figure 4c, that are promising for exhaustive column trays. They can achieve high gas and liquid performance with low hydraulic resistance of the tray (Figure 6, annular channels). They also ensure uniform distribution of gas in a liquid and a developed interphase surface on the tray.

It can be assumed that the lowest resistance of a ring swirler is due to the presence of one leading edge at the gas inlet to the channel, as opposed to channels with straight and profiled walls, as well as the difference in the gas velocity profile in each inlet device. The resistance of a profiled channel has a lower pressure drop compared to straight channels due to the difference in the slope of walls at the channel gas inlet. The slope angle in the channel with straight walls was 26° and in that with profiled ones 90°. We established that

an increase in the slope angle of the swirler channel walls leads to a decrease in the swirler channel resistance.

According to the obtained results, the magnitude of the resistance coefficient is subject to the dependence [32]:

$$\xi = A \times Re^{-k} \times l^{\,0.19} \times b^{0.6} \times h^{-p}, \tag{9}$$

where $u_G$ is the average gas flow rate in the swirler channels, m/s; $\rho_G$ is gas density, kg/m$^3$; $\mu$ is the dynamic viscosity coefficient of the gas, Pa $\times$ s; $l$ is the length of the swirler channel, m; $b$ is the width of the swirler channel, m; $h$ is the height of the swirler channel, m; $Re = (u_G \times b \times \rho_G)/\mu$ is the Reynolds number.

For the profiled channels (shown in Figure 4a) $A = 751$, $k = 0.28$, $p = 0.13$; the annular channels (Figure 4b) $A = 1571$, $k = 0.39$, $p = 0$.

The pressure drop of the trays with vortex devices is shown in Figure 7.

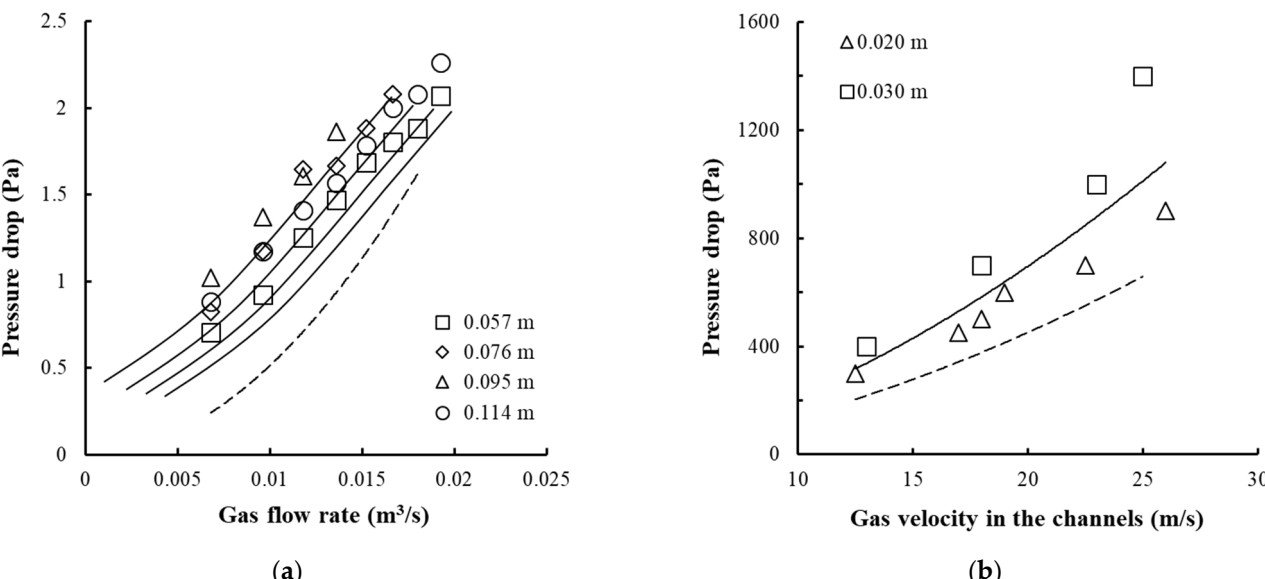

**Figure 7.** Change in pressure drop on the trays with vortex devices as a function of the gas flow rate (**a**) and the gas velocity in the channels (**b**). (**a**) Experimental points for the annular swirler shown in Figure 4c, for $R_{out} = 0.103$ m, $b = 0.005$ m, $b_{in} = 0.001$ m, $h = 0.008$ m, $n = 16$ pcs (1–4): for different liquid column height $H_0$, (**b**) Experimental points for the swirler shown in Figure 4b, for $R_{out} = 0.075$ m, $b = 0.005$ m, $h = 0.003$ m, $n = 40$ pcs (1–2): for different liquid column height $H_0$. The dashed lines represent dry tray resistance.

At a gas velocity of up to 35 m/s in the swirler channels and a liquid layer height of 0.1 m on the tray, its resistance does not exceed 2000 Pa.

### 3.2. Hydrodynamic Parameters of the Gas–Liquid Layer

The trays with vortex contact devices showed the following gas–liquid interaction modes: jet mode (Figure 8a), bubble mode (Figure 8b), bubble-annular mode (Figure 8c), annular mode. Due to high liquid loads, the bubble and bubble-annular modes are used for the exhaustive column trays.

Based on the previous numerical simulation results [20], the movement of the gas–liquid mixture on the tray of the exhaustive section of the column is represented as follows: gas jets leaving the swirler channels, having a high velocity head (unlike cap and valve contact devices), penetrate the liquid, break up, and create a localized bubble volume. Due to the gas phase energy, the resulting upward flow of bubbles entrains the liquid, forming a circulation loop of dimensions $D_{G-L}$. The circulating liquid captures and crushes the gas bubbles. This results in a developed interphase surface.

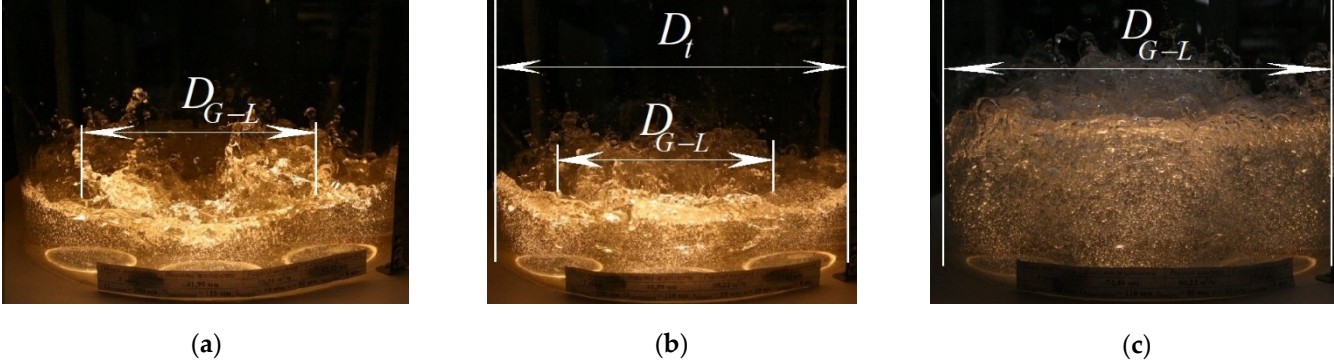

**Figure 8.** Structure of gas–liquid mixture on vortex tray: (**a**) jet mode; (**b**) bubble mode; (**c**) bubble-annular mode, $D_{G-L}$—gas–liquid diameter, m; $Dt$—diameter tray, m.

As the gas velocity in the swirler channels increases, the circulation of the liquid on the tray also increases. The experimental values of $D_{G-L}$ and the velocity of the gas–liquid medium in the circulation loop are shown in Figure 9.

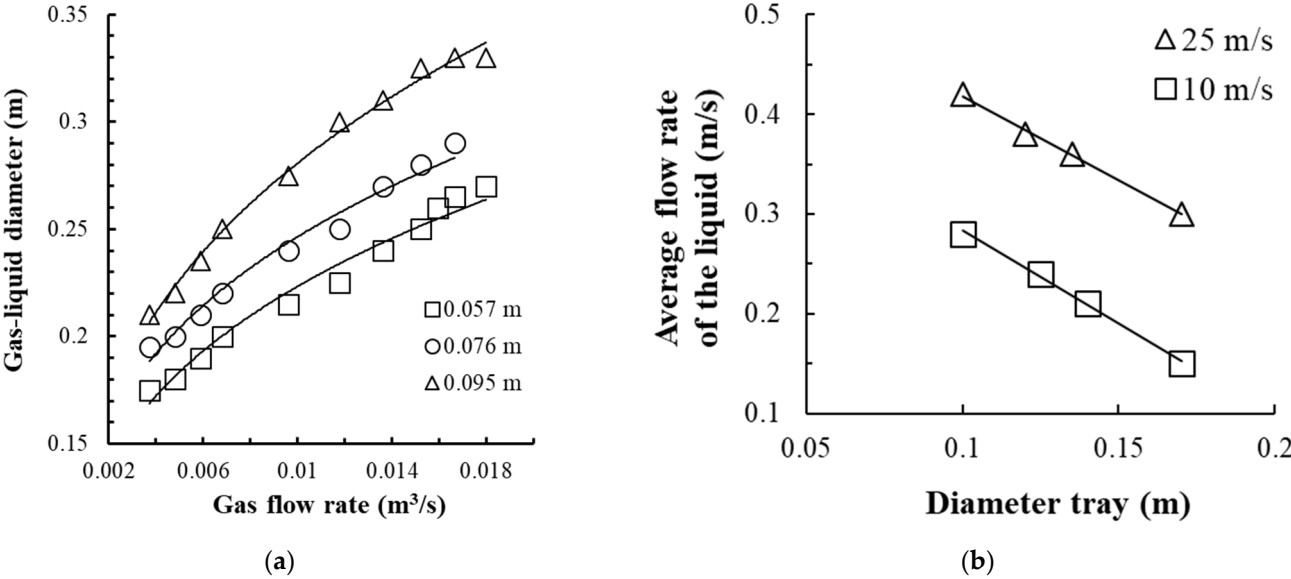

**Figure 9.** Change in the magnitude of the aerated liquid layer on the tray from the gas flow rate (**a**) and the average velocity of liquid movement on the tray from the tray diameter (**b**). (**a**) The tray with a contact device, shown in Figure 4c, at $R_{out}$ = 0.103 m, $h$ = 0.008 m, $b$ = 0.005 m, $b_{in}$ = 0.001 m, $n$ = 16 pcs. Experimental points (1–3): for different liquid column height $H_0$, (**b**) Experimental points (1–2): for different gas velocity $u_G$ in the swirler channels.

A transition from the bubble-annular mode to the annular mode of liquid flow on the tray is carried out when the gas velocity ratio is reached

$$u_G/u_{cr} \geq 1.25,$$

where $u_G$ is the average gas flow rate in the swirler channels; $u_{cr}$ is the critical gas velocity at which there is a transition from the bubble-annular mode to annular.

The critical gas velocity can be calculated according to the formula [33]:

$$\frac{u_{cr}}{0.27} = \left[ \frac{\rho_L \times (1 - \varphi) + \rho_G \times \varphi}{\rho_G} \times \frac{R_{out}^2}{r} \times \frac{\omega^2 \times V}{f \times \cos \alpha} \right]^{0.5}, \tag{10}$$

where $\rho_L$ is liquid density, kg/m$^3$; $\varphi$ is gas content; $\rho_G$ is gas density, kg/m$^3$; $R_{out}$ is the external swirler radius, m; $r$ is the cylindrical insert radius, m; $V$ is the volume of liquid in motion, m$^3$; $f$ is the cross-sectional area of the swirler's channels, m$^2$; $\omega$ is the liquid angular velocity, m/s; $\alpha$ is the swirler channel slope angle, deg.

The experimental values of the critical gas velocity are shown in Figure 10a, and the change in the liquid mass on the tray depending on the gas velocity is presented in Figure 10b.

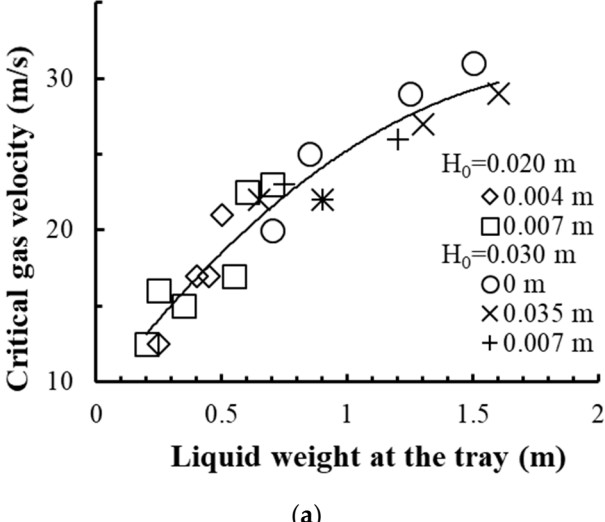
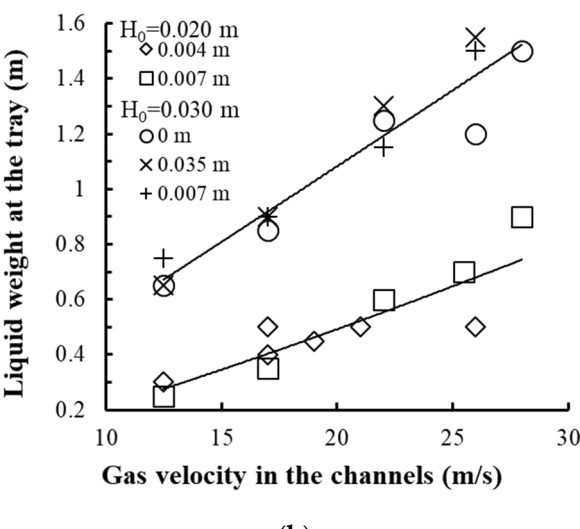

|  |  |
|:---:|:---:|
| (a) | (b) |

**Figure 10.** Dependence of the critical velocity on the liquid weight in the device (**a**) and the liquid weight on the vortex tray on the gas velocity in the swirler channels (**b**). Experimental data for the tray with the device are presented in Figure 4a at $R_{out}$ = 0.075 m, $b$ = 0.005 m, $h$ = 0.003 m, $n$ = 40 pcs. Experimental points (1–5): for different distance from the wall to the end of the drain bar $\delta$.

With an increase in the number of gas passage channels in the swirler and a decrease in the liquid weight in the device, the rotation of the gas–liquid layer is provided at a lower gas velocity in the channels and therefore at a lower pressure drop on the tray.

A preset liquid weight in the contact device is ensured by mounting the drain bar at a certain height above the tray deck and by the magnitude of gas (vapor) velocity in the swirler channels.

The liquid weight in the device increases with an increase in gas velocity in the swirler channels and from the drain bar mounting height. Changing the gap (distance) from the cylindrical insert wall to the drain bar $\delta$ from 0.001 to 0.009 m does not have a significant impact on the magnitude of the liquid weight on the tray.

The height of the rotating gas–liquid layer for these experimental conditions was 0.020–0.080 m.

The gas content on the tray for the bubble-annular and annular flow modes is presented in Figure 11a and can be calculated from the dependence [33].

The interphase surface of the gas–liquid layer on the vortex tray is shown in Figure 11b. In determining the interphase surface, the average surface diameter of the gas bubble $D_b$ was taken from experimental data and equaled 0.003–0.01 m$^{-1}$.

As established in the bubble and bubble-annular modes, the interphase surface increases with an increase in the swirler channel gas velocity. In the annular mode, it is approximately the same within the studied range of parameters. The maximum interphase surface was 1200 m$^{-1}$, which was comparable to the data achieved in apparatuses with turbine mixers.

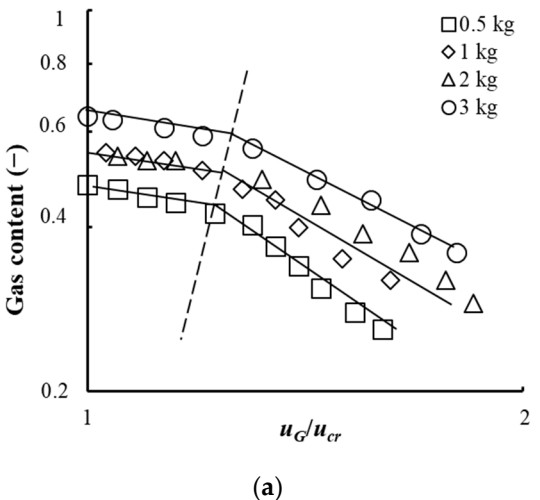
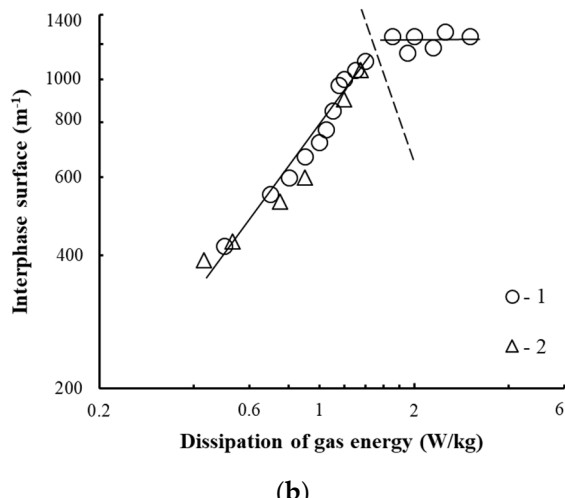

(**a**)                                (**b**)

**Figure 11.** Dependence of the gas content (**a**) on the ratio of gas velocities and the interphase surface on the gas energy dissipation (**b**). (**a**) The tray with contact devices shown in Figure 4b, at $n = 8$ pcs, $h = 0.008$ m, $R_{out} = 0.103$ m, $l_{chan} = 0.022$ m; $b = 0.003$ m. Experimental points (1–4): for different mass of liquid on the tray M. (**b**) The tray with contact devices shown in Figure 4b,c. Experimental points (1–3): 1—$R_{out} = 0.092$ m, $l = 0.022$ m; $b = 0.005$ m, $b_{in} = 0.001$ m, $n = 16$, 2—$R_{out} = 0.063$ m, $l = 0.011$ m; $b = 0.001$ m, $n = 8$ pcs. A dashed line represents a transition from the bubble-annular mode to annular.

## 4. Mass Transfer on the Vortex Tray

The experimental values of the mass transfer coefficient, in the context of absorption, obtained during the operation of vortex devices in the bubble-annular and annular modes are shown in Figure 12.

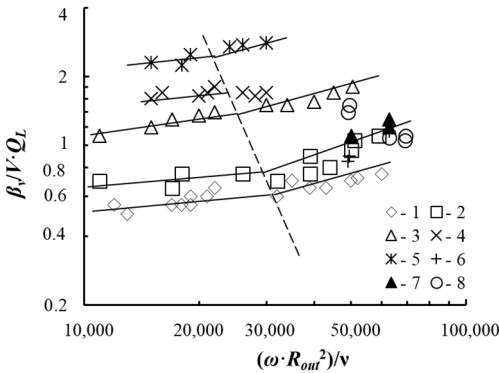

**Figure 12.** Change of the $\beta_V/V \times Q_L$ dimensionless parameter from the centrifugal Reynolds criterion. Experimental a (1–5) for the tray with contact devices are presented in Figure 4c at $Q_L = 0.8 \times 10^{-4}$—$6.1 \times 10^{-4}$ m$^3$/s, $R_{out} = 0.092$ m, $l = 0.022$ m; $b = 0.005$ m, $b_{in} = 0.001$ m, $n = 16$ pcs: 1—$H_0 = 0.034$ m; 2—$H_0 = 0.055$ m; 3—$H_0 = 0.07$ m; 4—$H_0 = 0.085$ m; 5—$H_0 = 0.1$ m. Experimental data (6–8) for the tray with contact devices are shown in Figure 4a at $H_0 = 0.03$ m, $\delta = 5 \dots 9$ mm: 6—$Q_L = 3.6 \times 10^{-6}$ m$^3$/s; 7—$Q_L = 13.6 \times 10^{-6}$ m$^3$/s; 8—$Q_L = 17.8 \times 10^{-6}$ m$^3$/s. The dashed line shows a change in modes on the tray.

The experimental data are consistent with calculated data (lines in Figure 11) according to the dependence [31]

$$\beta_V \times \frac{V}{Q_L} = A \times \left( \frac{\omega \times R_{out}^2}{\nu} \right)^p \times Sc^{0.5} \times \left( \frac{H_{G-L}}{h} \right)^{1.5}, \tag{11}$$

where $\beta_V$—mass transfer intensity, $s^{-1}$; $V$—volume of liquid in motion, $m^3$; $Q_L$—flow rate of oxygen-free water supplied to the tray, $m^3/s$; $A = 2 \times 10^{-3}$, $p = 0.24$ under bubble-annular flow mode; $A = 0.55 \times 10^{-3}$, $p = 0.38$—under annular flow mode; $\omega$—angular velocity of liquid, $s^{-1}$; $R_{out}$—external swirler radius, m; m; $v$—liquid kinematic viscosity coefficient, $m^2/s$; $H_{G\text{-}L}$—height of the gas–liquid layer, m; $h$—height of the channels in the swirler, m; $Sc$—Schmidt criterion.

The mass transfer coefficient ($h^{-1}$) can also be assessed according to the dependence proposed in the work [30]:

$$\beta_V = 136 \times \left[\varepsilon^{0.6} \times a^{0.8}\right]^{0.45}, \tag{12}$$

where a is the interphase surface, $m^{-1}$; $\varepsilon$ is the dissipation of gas energy, W/kg.

The angular velocity of the rotating gas–liquid layer on the cylindrical insert surface of the device was 10–20 $s^{-1}$.

The maximum dissipation of gas energy (up to 5 W/kg) was achieved on the vortex contact device with annular channels in the bubble-annular mode. In this case, the magnitude of the surface mass transfer coefficient was $0.5 \times 10^{-3}$ m/s.

## 5. Tray Efficiency

### 5.1. Vortex Tray Efficiency during Absorption

The efficiency values of the tray with vortex contact devices presented in (Figure 4b,c) are shown in Figure 13.

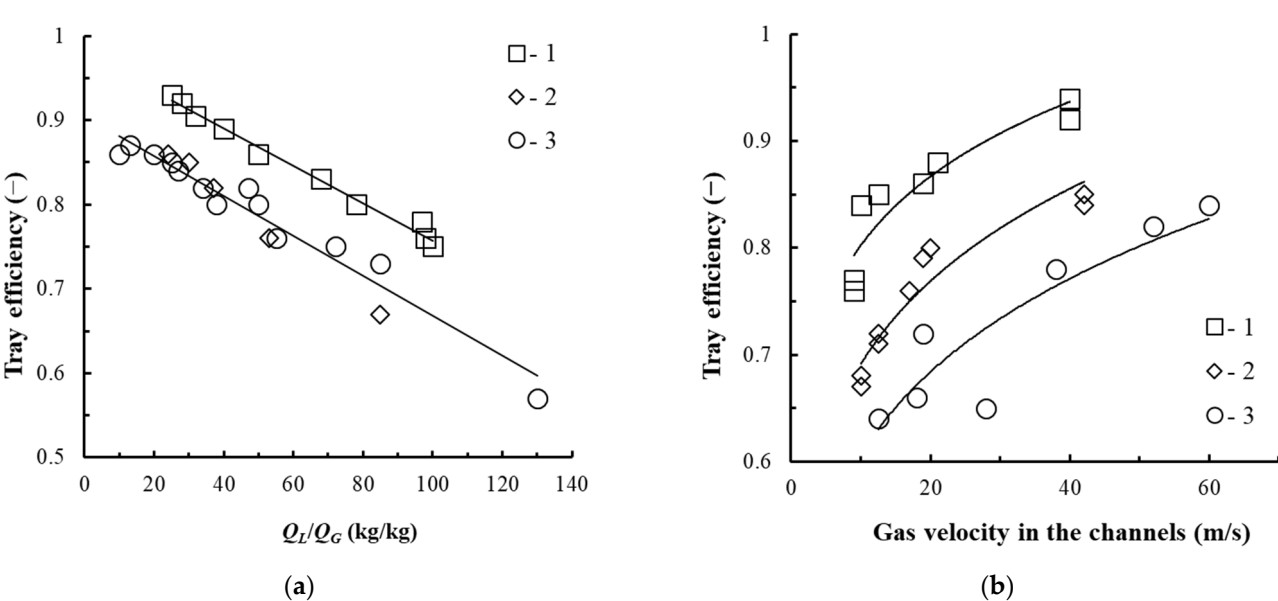

**Figure 13.** Dependence of the tray efficiency on the ratio of flow rates (**a**) and the gas velocity in the swirler channels (**b**). Experimental points for the annular swirlers at $h = 0.008$ m, $Q_L = 0.000275$–$0.000725$ $m^3/s$ (1–3): 1—swirler shown in Figure 4c, at $R_{out} = 0.092$ m, $l = 0.022$ m, $b = 0.005$ m, $b_{in} = 0.001$ m, $n = 16$ pcs; 2—swirler shown in Figure 4b, at $R_{out} = 0.103$ m, $l_{chan} = 0.011$ m; $b = 0.005$ m, $n = 8$ pcs; 3—swirler shown in Figure 4a, $l = 0.022$ m; $b = 0.004$ m, $R_{out} = 0.065$ m, $n = 8$ pcs.

As established, with an increase in the gas velocity in the swirler channels and a decrease in the flow rate of the liquid on the tray, its efficiency increases. According to [21], at relatively low concentrations of ethanol in the liquid (x < 5% wt), when the main resistance to the mass transfer of the ethanol–water mixture is concentrated in the liquid phase, the efficiency of the tray with vortex contact devices in adiabatic rectification and in physical absorption in the bubble-annular mode is almost the same.

In this regard, for the trays with vortex devices (see Figure 4b) intended for the exhaustive section of the rectification alcohol column where the ethanol concentration is relatively low, their efficiency can be taken to be 0.7.

### 5.2. Vortex Tray Efficiency during Rectification

According to the data presented in Figure 14. The tray efficiency in adiabatic rectification (Point 1 in Figure 14) was equal to 0.4–0.6. The tray efficiency increases with a decrease in ethanol concentration in the mixture and does not significantly depend on the steam velocity in the swirler channels, which is consistent with the data [21], (Point 5, in Figure 14).

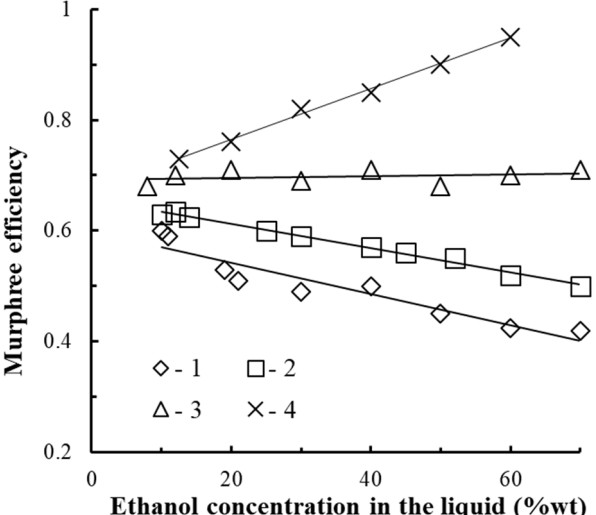

**Figure 14.** Change in the efficiency of the $E$ vortex tray from ethanol concentration in the $x$ liquid at $u_{vap}$ = 15–25 m/s. Experimental points (1–4): 1—adiabatic rectification; 2–4—diabatic rectification; 2—$Q_{vap}/Q_{con}$ = 25; 3—4; 4—1.8.

When the steam velocity in the swirler channels varies from 15 to 25 m/s, the tray efficiency does not change significantly.

The tray efficiency in diabatic rectification (Points 2–4 in Figure 14) was 0.6–0.95 and depends on the $Q_{vap}/Q_{con}$ flow ratio on the tray. The higher the flow rate of condensing vapors on the surface of the dephlegmator heat exchange pipe, the higher the tray efficiency.

The increase in the tray efficiency compared to adiabatic rectification is due to the effect caused by partial rectification [34,35]. That is, when condensing from the vapor phase, vapors of the high boiling component fall out in greater quantities than the easily volatile component, which causes the vapors of the ascending mixture to strengthen along the trays.

As studies have shown (see Figure 15), in partial rectification, an increase in the $Q_{vap}/Q_{con}$ flow ratio results in an increase in the concentration of the easily volatile component in the vapors.

Application of diabatic rectification on the trays with vortex contact devices allows intensifying the mass transfer in the vapor phase.

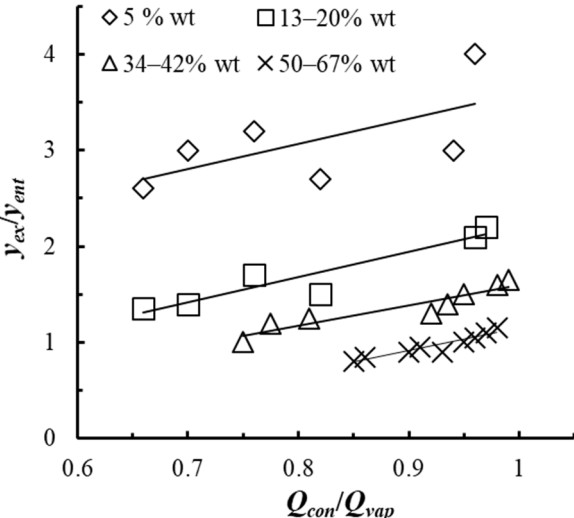

**Figure 15.** Dependence of the $y_{ex}/y_{ent}$ concentration ratio in the vapor mixture on the $Q_{vap}/Q_{con}$ ratio. Experimental points: (1–4): for different ethanol concentration in the liquid on the tray $x$.

## 6. Diabatic Alcohol Columns

### 6.1. Column Design

A design of an alcohol diabatic column was created based on the developed and studied vortex devices. The diagram is presented in Figure 16.

The column consists of lower exhaustive column 1, upper concentrating column 2 equipped with chambers 3 and 4 for heat-transfer medium outlet, including steam passage pipe connectors 5.

Vortex contact devices 7 (shown in Figure 4a) are located in the concentrating section of column 2 on trays 6. Heat exchange pipes 8 are also fitted to create a reflux in each vortex device due to partial condensation of the mixture ascending vapors on their outer surface. The lower end of the heat exchange pipes is connected to chamber 4 equipped with connecting piece 9 for venting gas into the vacuum line and connecting piece 10 for removing the heat-transfer medium. The upper end of the heat exchange pipes is connected to the cavity of chamber 3 where the heat-transfer medium is supplied through connecting piece 11 and is equipped with pipe connectors for creating liquid film 12.

In the exhaustive section of column 1 on trays 13, vortex devices 14 (shown in Figure 4c) are fitted to operate in the bubble and bubble-annular modes.

The tray of the exhaustive column includes liquid flows 15, hydraulic lock cup 16, and vortex devices 14.

Vortex devices 7 are fastened and cylindrical inserts 17 are located on tray 6 of the concentrating column along concentric circles.

Cavity 20 allows the trays to be assembled and disassembled in the shells of column 2. The cylindrical inserts installed tightly against the tray prevents the ingress of liquid between them, which eliminates stagnant zones and reduces the volume of liquid in the device.

The column is also equipped with process pipe connectors and bosses for the process and fraction selection.

During operation of the column, vapors from the still pass through the vortex contact devices 14 located on the trays of the column exhaustive section and have contact with the liquid on it. After the exhaustive section of the column, the vapors pass through pipe connector 5 and enter vortex contact devices 7 located on trays 6 of the concentrating section of the column. There, they interact with the reflux rotating in the form of a gas–liquid layer on the inner surface of cylindrical inserts 17. The reflux forms on each tray due to condensation of the vapor on the surface of the heat exchange pipes, which then flows from one tray to another through the overflow pipe connectors. The heat-transfer medium (for

example, water) is supplied to chamber 3 through connecting piece 11, which then enters the annular gap formed by pipe connector 12 and pipe 8, and there forms a film. The liquid film flows down the inner surface of heat exchange pipes 8, providing heat removal (vapor condensation). At the heat exchange pipe outlet, water flows into chamber 4 and exits the column through connecting piece 10. To ensure the boiling of the gravity-drainage water film in the cavity of pipes 8, a vacuum is provided in chambers 3 and 4 by connecting the vacuum line to connecting piece 9.

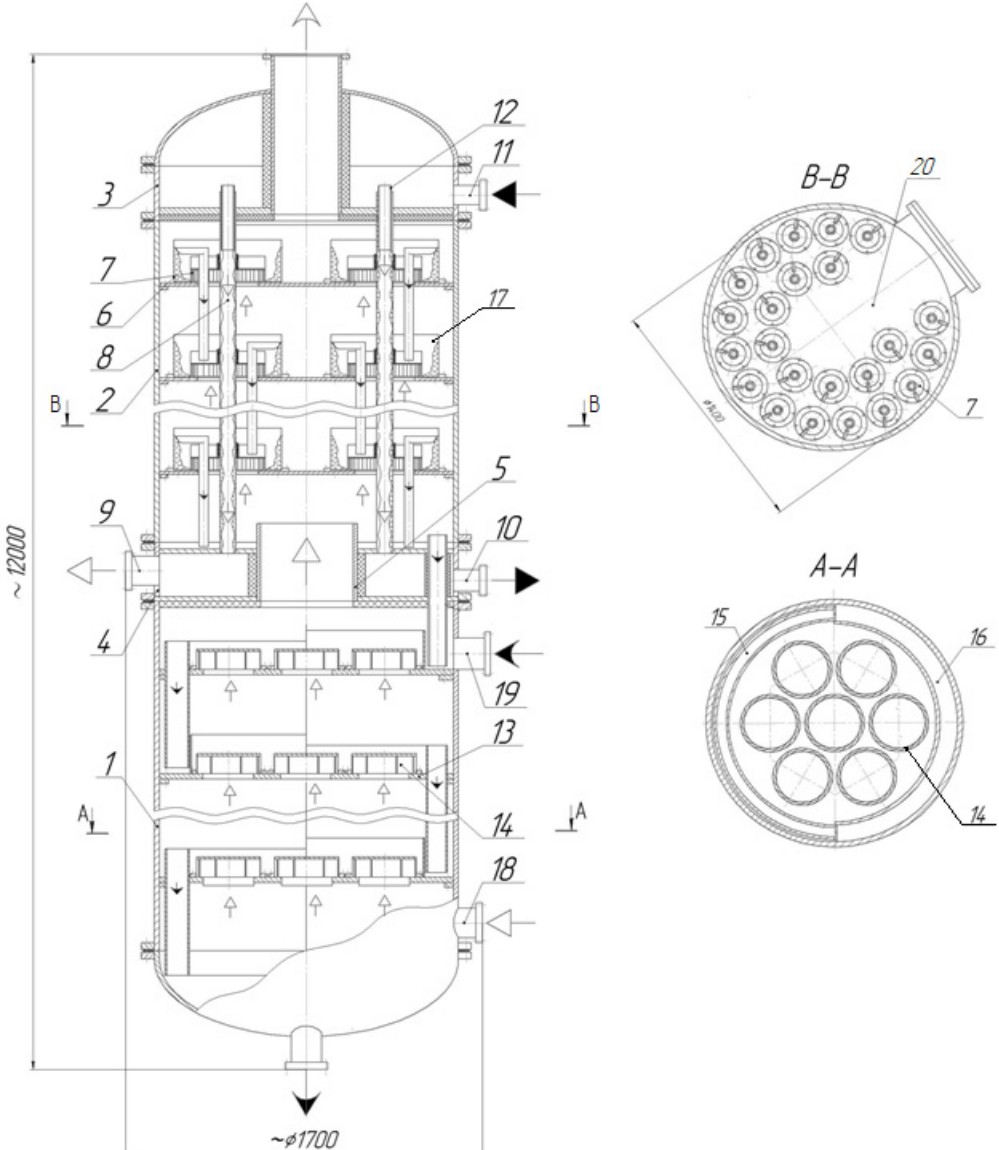

**Figure 16.** Diagram of the diabatic alcohol column. 1—exhaustive column; 2—concentrating column; 3 and 4—heat-transfer medium outlet and inlet chamber; 5—steam pipe connector; 6—tray; 7—vortex contact device; 8—dephlegmator heat exchange pipes; 9—gas drain connecting piece; 10—heat-transfer medium injection connecting piece; 11—heat-transfer medium supply connecting piece; 12—liquid film pipe connector; 13—lower tray; 14—vortex devices; 15—overflows; 16—hydraulic lock cup; 17—cylindrical insert; 18, 19—connecting pieces, 20—cavity. ➤—heat-transfer medium; ▷—steam; ➤—working mixture.

At an ethanol column capacity of 2900 L/h and a reflux ratio of four, the vapor flow rate of the column concentrating section was equal to 2.2 m$^3$/s, while the total reflux flow rate was 2.48 kg/s. The column diameter was 1.4 m.

In this case, the number of vortex contact devices on the concentrating column trays was 23 pieces. The cylindrical insert diameter was 0.2 m, the swirler diameter was 0.17 m, the number of channels in the swirler was 40 pcs, the channel height was 0.008–0.01 m, and the channel width was 0.005–0.01 m. The diameter of heat exchange pipes was 0.03 m.

The number of vortex devices on the exhaustive column trays was 7 pieces, their diameter 0.35 m, and the internal 0.25 m. The channel height was 0.045 m, the channel width 0.011 m, and the number of channels 11 pcs.

### 6.2. Comparing the Alcohol Column Parameters at Ethanol Capacity 2900 L/h in Adiabatic and Diabatic Rectification

Height of column concentrating sections. With the accepted theoretical number of concentrating trays 25 pcs. [36], the number of actual vortex trays in adiabatic rectification and an efficiency of 0.4 was 63 pcs. At an efficiency of 0.7, the number of trays in the diabatic column with vortex devices was 36 pcs. With the accepted distance between the trays of 300 mm, the height of the working section of the adiabatic concentrating column would be 19 m, and that of the diabatic concentrating column would be 11 m.

Dephlegmator surface. The magnitude of the heat transfer coefficient in the pipes of the multiway shell-and-tube dephlegmator of the adiabatic alcohol column fitted in the upper part of the unit according to the industrial data [36] is 580 W/(m$^2$ K). With different designs of the dephlegmator and the varying temperature of the reflux entering the upper tray, the surface of the adiabatic column dephlegmator could be [36] 100–185 m$^2$.

In the diabatic column, the dephlegmator consists of heat exchange pipes built into the column height. Heat removal in the dephlegmator pipes is carried out due to the boiling of a liquid film caused by reduced pressure in the heat exchange pipe cavity. According to the data [26], the heat transfer coefficient during boiling in the film is 4000 W/(m$^2$ K). During condensation of the ethanol–water mixture vapors [36], the heat transfer coefficient is 2000 W/(m$^2$ K). Then the heat exchange surface of the heat exchange pipes is 72 m$^2$. This is lower than that of the adiabatic column.

It is important to note that the heat transfer coefficient during vapor condensation in the adiabatic column dephlegmator is lower than that on the heat exchange pipe surface in the diabatic column. This is due to the relatively low thermophysical parameters of condensed vapors in the upper section of the adiabatic column (a small magnitude of a phase transition criterion) [26]. This results in an increase in the heat exchange surface in the adiabatic column dephlegmator.

Heat-transfer medium consumption. In the shell-and-tube dephlegmator of the adiabatic alcohol column, based on the thermal balance, when heating water from 20 to 70 °C, the water flow rate was 60 m$^3$/h.

Heat removal in the diabatic column is carried out in a draining water film during boiling and evaporation of vapors from its surface. Heat removal takes place mainly due to a phase transition. In this case, the flow rate of water supplied to the heat exchange pipes is only necessary to maintain a turbulent mode in the draining film. Therefore, the water flow rate is three times lower than the adiabatic column (20 m$^3$/h). This heat-transfer medium flow rate ensures the necessary Reynolds number to maintain the heat transfer coefficient [24] in the film during boiling.

Energy consumption. The temperature of the condensate flowing from the multi-pass dephlegmator pipe surface onto the upper tray of the adiabatic column was 30–60 °C. Therefore, an additional heat flow of the ascending steam was spent on heating the condensate to the boiling point. On average, according to [36], a difference between the incoming reflux temperature and the rising vapor temperature is 10 °C. Then the heat flow of vapors spent on heating the reflux on the upper tray is ~104 kW.

According to experimental data, in a diabatic alcohol column, the supercooling of condensate on the dephlegmator pipe surface will be maximum 3 °C. This results in a decrease in the flow rate of rising vapors in the column for heating the condensate.

It is important to note that in a diabatic column, the heat-transfer medium is reused to irrigate the dephlegmator pipes without a preliminary cooling procedure, for example, in a cooling tower. This also results in a decrease in energy consumption.

During rectification under pressure in an alcoholic diabatic column, it is possible to use secondary steam produced in the cavity of the heat exchange pipes during the boiling of the heat-transfer medium. This, when installing the heat pump (ejector), will save up to 30% steam.

There are possible options for reducing energy consumption when replacing the dephlegmator pipes with heat ones [37–39].

Summing up the results presented in the article, the number of trays in the diabatic column decreased from 63 to 36 pieces and the height of the unit decreased by half compared to the adiabatic one. With a column diameter of 1.4 m, there were 7–23 vortex contact devices on the tray, whereas in the adiabatic column with the cap trays, there were 49. The surface of the reflux condenser heat exchange pipes of the diabatic column was 72 m$^2$, while that of the adiabatic column was 100–185 m$^2$.

The flow rate of water supplied to the reflux condenser pipes decreased by three times and amounted to 20 m$^3$/h. The heat flow of vapors spent on heating the reflux in the diabatic column decreased three times, and the volume of liquid on the tray deck of the concentrating section of the columns decreased six times. Additional measures specified in the article, for example, the installation of a heat pump (ejector), will save up to 30% of the steam supplied to the column.

## 7. Conclusions

Based on experimental data obtained from studying hydrodynamics, mass transfer, and the rectification process on single-element trays, we developed a new design of an alcohol diabatic distillation column with built-in heat exchange pipes as the reflux condenser. The column is fitted with two types of vortex contact devices designed for the concentrating and exhaustive sections of the column.

The new contact devices for the column concentrating section allow reducing the volume of liquid on the tray deck, eliminating stagnant zones, evenly distributing the reflux, and decreasing the flow rate of steam and the heat-transfer medium. At that, the efficiency increases due to the partial rectification and intensification of the mass and heat flow transfer.

The new tray design for the column exhaustive section allows eliminating the fluctuation of liquid on the tray deck due to the uniform distribution of steam jets in the liquid, increasing steam energy dissipation, and evenly distributing steam bubbles and organizing their circulation in the liquid layer on the tray. The design provides an increase in the steam load and the supplied liquid, allows organizing a large interphase surface of up to 1200 m$^{-1}$, and increases the mass-transfer coefficient up to $(0.5–0.8) \times 10^{-3}$ m/s.

The dependencies were determined and tested. They allow determining the tray operating mode, gas content, hydraulic resistance, interphase surface magnitude, angular velocity of the liquid–vapor mixture, magnitude of the mass-transfer coefficient, and efficiency.

The results show the prospects of using new vortex contact devices and trays, as well as the proposed design of the diabatic column.

**Author Contributions:** N.A.V. conceived and designed the experiments. A.V.B. performed the majority of experiments and D.A.Z. performed additional experiments. All authors contributed to the data analysis and writing of the paper. All authors have read and agreed to the published version of the manuscript.

**Funding:** The work reported in paper «Intensification of Heat and Mass Transfer in a Diabatic Column with Vortex Trays» was supported by the Ministry of Science and Higher Education of the Russian Federation within the framework of State Assignment of the "Technology and equipment for the chemical processing of biomass of plant raw materials" project FEFE-2020 0016.

**Institutional Review Board Statement:** Not applicable.

**Informed Consent Statement:** Not applicable.

**Data Availability Statement:** Not applicable.

**Conflicts of Interest:** The authors declare no conflict of interest.

## Abbreviations

| | |
|---|---|
| $A$ | coefficient |
| $a$ | interphase surface ($m^{-1}$) |
| $b$ | width of the channel in the swirler (m) |
| $b_{in}$ | internal width of the channel in the swirler (m) |
| $c$ | oxygen concentration in the liquid on the tray ($kg/m^3$) |
| $c_0$ | concentration of oxygen in the liquid supplied to the tray ($kg/m^3$) |
| $c^*$ | equilibrium concentration of oxygen in the liquid ($kg/m^3$) |
| $D_b$ | average surface diameter of the bubble (m) |
| $D_{G\text{-}L}$ | gas–liquid diameter (m) |
| $Dt$ | diameter tray, m |
| $D_{in}$ | inner diameter of the swirler (m) |
| $E$ | Murphree efficiency |
| $f$ | cross-sectional area of the swirler's channels ($m^2$) |
| $G$ | vapor phase flow rate (kg/s) |
| $h$ | swirler channel height (m) |
| $H_0$ | liquid level at the stage (m) |
| $H_{G\text{-}L}$ | liquid height on the plate (m) |
| $k, p$ | degree |
| $L$ | liquid phase flow rate (kg/s) |
| $l$ | swirler channel length (m) |
| $M$ | liquid weight (kg) |
| $m$ | slope of the equilibrium curve of the mixture |
| $n$ | number of channels (pcs) |
| $Q_{con}$ | condensate flow rate ($m^3/s$) |
| $Q_G$ | gas flow rate ($m^3/s$) |
| $Q_L$ | flow rate of oxygen-free water supplied to the tray ($m^3/s$) |
| $Q_{vap}$ | steam flow rate ($m^3/s$) |
| $r$ | radius of the cylindrical insert of the vortex device (m) |
| $Re$ | Reynolds number of gas |
| $R_{in}$ | inner radius swirler (m) |
| $R_{out}$ | external swirler radius (m) |
| $Sc$ | Schmidt criterion |
| $U$ | internal energy of the gas (W) |
| $u_{cr}$ | critical gas velocity (m/s) |
| $u_G$ | average flow rate of the gas in the swirler channels (m/s) |
| $u_L$ | average flow rate of the liquid (m/s) |
| $u_{vap}$ | vapour velocity (m/s) |
| $V$ | volume of liquid on the tray ($m^3$) |
| $x$ | ethanol concentration in the liquid on the tray (% wt) |
| $y_{ent}$ | concentration of ethanol vapor at the tray inlet (% wt) |

| $y_{ex}$ | concentration of ethanol vapor at the outlet of the tray (% wt) |
| $y^*$ | equilibrium concentration of vapors with a liquid placed on the tray (% wt) |
| $\alpha$ | angle of the wall slope of the swirler's channel (deg) |
| $\beta_V$ | mass transfer intensity ($s^{-1}$) |
| $\delta$ | distance from the wall to drain bar (m) |
| $\Delta P$ | pressure drop (Pa) |
| $\varepsilon$ | dissipation of gas energy (W/kg) |
| $\eta$ | tray efficiency |
| $\mu$ | coefficient of dynamic viscosity of the gas (Pa×s) |
| $\nu$ | coefficient of kinematic viscosity of the liquid ($m^2/s$) |
| $\xi$ | resistance coefficient |
| $\rho_G$ | gas density ($kg/m^3$) |
| $\rho_L$ | liquid density ($kg/m^3$) |
| $\varphi$ | gas content |
| $\omega$ | angular velocity of the liquid ($s^{-1}$) |

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
