# Peer review of "Intensification of Heat and Mass Transfer in a Diabatic Column with Vortex Trays"

_2305-7084, doi:10.3390/chemengineering6020029_

Round 1

Reviewer 1 Report

The paper developed a new design of an alcohol diabatic column with heat exchange pipes (as a dephlegmator) passed through the trays and demonstrated the steam consumption. Generally the paper is not too bad. However, there are several issues needs to be corrected before the paper can be further considered for second review. Detailed information are as follows:

  1. The abstract is not good, please revise it including the main findings.
  2. The language is not good as some descriptions are informal, such as “a heat and mass transfer...”
  3. The logical of the introduction section is not constructive, please revise. It is advised that the novelty of the present study should be emphasized.   
  4. The format of the manuscript is bad, such as equations and figures, please revise.
  5. Some figures are not clear enough, please revise.  
  6. The superiority of the design should be summarized clearly.
  7. Again, the conclusion sectionis not constructive, the main findings resulting from the study is not clearly identified.

Author Response

Thanks for the substantive comments. The authors answers are presented below.

1. The abstract is not good, please revise it including the main findings.

The authors revised the abstract to the article and presented it as:

We used the vortex contact devices that we developed and investigated to make a new design of an alcohol diabatic distillation column with heat exchange pipes (as the reflux condenser) pass-ing through the concentrating section trays. In the column, ascending vapors partially condense on the surface of vertically installed heat exchange tubes, forming a reflux. The reflux is then mixed with the draining liquid flow in the vortex contact devices placed on the trays. Heat is removed from the column through the boiling of the draining water film along the inner surface of the heat exchange pipes.

 We compared both diabatic and adiabatic columns fitted with the developed vortex con-tact devices on the trays. The proposed innovative contact system allows increasing productivi-ty, reducing column dimensions and steam and heat-transfer medium consumption, and in-creasing separation efficiency. 

Dependences for calculating the gas content, hydraulic resistance, and interphase surface re-quired for designing the vortex contact devices of the proposed unit trays are presented.

2. The language is not good as some descriptions are informal, such as “a heat and mass transfer...”

The analysis of the material of the corrected article is carried out, informal descriptions are excluded if possible

3. The logical of the introduction section is not constructive, please revise. It is advised that the novelty of the present study should be emphasized.

The authors have revised part of the introduction of the article, and supplemented it with the following content:

Rectifying columns are widely used in industry to separate mixtures and make products for different purposes. The adiabatic units that are currently used have high energy consumption, which results in increased rectification costs. Rectifying columns have a height often exceeding 25 meters. All this leads to high costs related to the manufacture, operation, and repair of rectifying columns.

The method used to organize refluxes in the adiabatic units does not provide optimal distribution of mixture components both on the trays and along the column height. The steam flow passes through the column and is condensed in the heat exchangers (reflux condenser). The resulting condensate enters the upper column trays with a maximum flow. This leads to several factors that reduce unit efficiency. In particular, with this reflux formation, the intensity of vapor condensation on the surface of heat exchanger pipes decreases due to the relatively low thermophysical parameters of the condensed vapors produced in the upper part of the adiabatic column (low phase transition criterion). This leads to an unjustified increase in the heat exchange surface, the dimensions of the heat exchange devices, high metal consumption, and increased consumption of heat-transfer media. Reuse of the heat-transfer medium passed through the reflux condenser requires cooling, which also results in additional energy consumption.

The heat and mass exchange processes (condensation and evaporation) on the trays along the column height are not synchronized between the mixture low-boiling and high-boiling components due to the difference in the physical properties of the draining maximum reflux flow and rising vapors.  This suggests the practicability of creating a reflux on each tray, for example, by partial condensation of vapors rising along the column height.

Additionally, in the adiabatic column, we observe the supercooling of the reflux flow on the surface of multi-pass heat exchangers, despite the constant improvement of the reflux condenser design. It is obvious that a decrease in the heat-transfer medium flow rate triggers a laminar flow in the pipes, which is not permissible, and its increase leads to the pipe surface and condensate cooling. Use of different heat exchange intensifiers in multi-pass reflux condensers leads to an increase in dimensions and metal consumption. Reflux supercooling requires additional heating to the boiling point. The reflux not heated to the boiling point condenses volatile components (impurities) on the upper trays and entrains them to the bottom of the column. It is therefore not possible to completely separate and purify a distillate from impurities, which is currently in demand.

An unreasonably large flow of the reflux on the upper trays leads to an increase in the diameter of the concentrating section of the column to accommodate overall overflows on them and increase the number of contact devices.

The displacement of a large vapor flow along the column results in increased unit resistance and heat losses.

The use of conventional contact devices on trays such as fixed and movable valves or caps is justified only at low unit capacity. An increase in the liquid flow rate leads to a sharp decrease in the tray efficiency due to the flow of gas bubbles from the gas-liquid layer.  Additionally, an increase in the steam load on the trays with such contact devices leads to liquid pulsations, drop entrainment, and flooding. Stagnant zones form on the trays, which leads to the formation of deposits on their surfaces.

With conventional trays, a large volume of liquid is typically placed to provide a certain height of the liquid layer, which keeps them from being used, for example, for processing explosive and thermolabile mixtures.  This also keeps the unit from being quickly brought to its operating mode.

The above factors prove that new design solutions are required both in the creation of rectifying columns and in the choice of contact devices on the tray.

And we also emphasized the novelty of the presented studies and are shown in the corrected version of the article.

This paper presents work on the development of a diabatic column design employing a new approach to reflux formation on trays and the organization of methods for intensifying heat removal and separation, which allows reducing capital and operating costs due to a decrease in unit dimensions and steam and heat-transfer medium consumption.

We developed new designs of contact vortex devices on trays, which allowed lowering the volume of liquid on the tray deck, reducing sediment formation conditions and the number of stagnant zones, and increasing tray efficiency and steam and liquid productivity.

4. The format of the manuscript is bad, such as equations and figures, please revise.

The format of the drawings and captions has been corrected and presented in the corrected version of the article

5. Some figures are not clear enough, please revise.

The drawings have been revised.

6. The superiority of the design should be summarized clearly.

After comparing the adiabatic and diabatic columns, the superiority of the diabatic column is shown in the form of:

Summing up the results presented in the article, the number of trays in the diabatic column decreased from 63 to 36 pieces and the height of the unit decreased by half com-pared to the adiabatic one. With a column diameter of 1.4 m, there were 7 - 23 vortex con-tact devices on the tray, whereas in the adiabatic column with the cap trays, there were 49. The surface of the reflux condenser heat exchange pipes of the diabatic column was 72 m2, while that of the adiabatic column was 100-185 m2.

The flow rate of water supplied to the reflux condenser pipes decreased by three times and amounted to 20 m3/h. The heat flow of vapors spent on heating the reflux in the dia-batic column decreased three times, and the volume of liquid on the tray deck of the con-centrating section of the columns decreased 6 times.  Additional measures specified in the article, for example, the installation of a heat pump (ejector), will save up to 30% of the steam supplied to the column.

7. Again, the conclusion sectionis not constructive, the main findings resulting from the study is not clearly identified.

The conclusion has been revised and presented in the form:

Based on experimental data obtained from studying hydrodynamics, mass transfer, and the rectification process on single-element trays, we developed a new design of an al-cohol diabatic distillation column with built-in heat exchange pipes as the reflux conden-ser. The column is fitted with two types of vortex contact devices designed for the concen-trating and exhaustive sections of the column. 

The new contact devices for the column concentrating section allow reducing the volume of liquid on the tray deck, eliminating stagnant zones, evenly distributing the re-flux, and decreasing the flow rate of steam and the heat-transfer medium. At that, the effi-ciency increases due to the partial rectification and intensification of the mass and heat flow transfer.

The new tray design for the column exhaustive section allows eliminating the fluctu-ation of liquid on the tray deck due to the uniform distribution of steam jets in the liquid, increasing steam energy dissipation, and evenly distributing steam bubbles and organiz-ing their circulation in the liquid layer on the tray. The design provides an increase in the steam load and the supplied liquid, allows organizing a large interphase surface of up to 1,200 m-1, and increases the mass-transfer coefficient up to (0.5 - 0.8)·10-3 m/s.

The dependencies were determined and tested. They allow determining the tray op-erating mode, gas content, hydraulic resistance, interphase surface magnitude, angular velocity of the liquid-vapor mixture, magnitude of the mass-transfer coefficient, and effi-ciency.

The results show the prospects of using new vortex contact devices and trays, as well as the proposed design of the diabatic column.

Reviewer 2 Report

This work is very original, and is focused on process intensification now keywords in chemical and process engineering and definitely interesting for the readers of this Journal.
The authors presented an innovative contact system, improving the performance of commonly used trays in the column.
The results are quite convincing and the conclusions are supported by case study addressed.
The following are some minor interventions to recommend to the authors: 

1. In general, the column diameter affects the fluid dynamics of the process; considering that the contact unit is a plate with particular elements and the column diameter is 0.1 m, what could be the effect of the diameter on the results obtained? Do you expect an improvement or deterioration in overall performance?

2. Could you add some references to Eq. 1, and better describe the application range and approximations to be made? 

3. In Eq 8, shouldn't the concentrations be expressed in moles according to Murphree?

4. Could you add discussion of Figure 6?  

5. Describe the parameters used in Figure 8, Dt, DG-L, in the caption.

6. In Figuara 10, why did you use the mass of liquid, and not its flow rate?

7. In Figure 12, the markers numbered 1-7, what are they? Please indicate what they are. Also for figure 13 and 14

8. On page 14, I think there is a sentence in Cyrillic.

9. Fig. 14, I think it should be x molar, could you check the textbooks and verify the Murphree equation?

- It would also be nice to see some video of the equipment in operation, if that is possible.

Author Response

Thanks for the substantive comments. The authors answers are presented below.

1. In general, the column diameter affects the fluid dynamics of the process; considering that the contact unit is a plate with particular elements and the column diameter is 0.1 m, what could be the effect of the diameter on the results obtained? Do you expect an improvement or deterioration in overall performance?

An increase in the diameter of the column and, consequently, the number of contact devices affects the uniformity of irrigation of the tray plate of the exhaustive part of the column. But due to the fact that the developed vortex contact devices provide intensive mixing of the liquid, due to the high energy dissipation of gas jets coming out of the channels of the swirler, the effect of the liquid flow on efficiency will not be significant. As for the plates of the reinforcing part of the column. In this case, the uniformity of irrigation does not depend on the diameter of the column. Since the phlegm is created on the plate not by entering the flow, but due to partial condensation of steam on the heat exchange pipes of the deflegmator passed through the plates.

2. Could you add some references to Eq. 1, and better describe the application range and approximations to be made?

The authors indicated in the materials of the corrected article a link [21] to the source where the formula (1) is presented. Initially, this dependence was obtained by the authors of the article and presented in the dissertation of D.A. Zemtsov, Ph.D., on the topic "development of a thermal rectification column in plant raw materials processing technologies". The formula is obtained on the basis of experimental data by separating the ethanol-water mixture into a vortex adiabatic column with a diameter of 200 mm with a number of plates of 25 pcs.

3. In Eq 8, shouldn't the concentrations be expressed in moles according to Murphree?

Really, the efficiency was calculated using molar concentrations. The authors corrected an annoying mistake made when writing the notation in equation (8).

4. Could you add discussion of Figure 6?

The authors added a discussion in the form:

It can be assumed that the lowest resistance of a ring swirler is due to the presence of one leading edge at the gas inlet to the channel, as opposed to channels with straight and profiled walls, as well as the difference in the gas velocity profile in each inlet device. The resistance of a profiled channel has a lower pressure drop compared to straight channels due to the difference in the slope of walls at the channel gas inlet. The slope angle in the channel with straight walls was 26° and in that with profiled ones 90°. We established that an increase in the slope angle of the swirler channel walls leads to a decrease in the swirler channel resistance.

5. Describe the parameters used in Figure 8, Dt, DG-L, in the caption.

The designation of the parameters: DG-L – gas-liquid diameter (m); Dt – diameter tray (m) shown in the caption Fig.8, and also presented in the notation of the corrected version of the article.

6. In Figuara 10, why did you use the mass of liquid, and not its flow rate?

The liquid flow rate does not significantly affect the mass of the rotating the liquid layer on the surface of the cylindrical insert. The incoming liquid to the contact device updates this layer, and then flows down the drain strips from plate to plate. The mass of the liquid in the device depends on the design parameters of the dish, to a greater extent on the height of the installation of the drain bar Ho and the flow rate of steam (gas)

7. In Figure 12, the markers numbered 1-7, what are they? Please indicate what they are. Also for figure 13 and 14

The design of the drawings has been corrected and the positions of the numbers have been decrypted

8. On page 14, I think there is a sentence in Cyrillic.

The typo on page 14 has been removed.

9. Fig. 14, I think it should be x molar, could you check the textbooks and verify the Murphree equation?

Really, the Murphrey efficiency is calculated by us in the presented manuscript in molar fractions and is shown on the graph on the ordinate axis. The authors' representation of mass concentrations along the abscissa axis does not distort the qualitative representation of the graph, but makes it easier for the reader to understand

10. It would also be nice to see some video of the equipment in operation, if that is possible.

Also in the updated work, the section annotations, introduction, conclusions has been revised and the novelty of the work is shown.

Round 2

Reviewer 1 Report

The manuscript can be accepted now.